# Representation Learning of Ancient Greek Letterforms across Time

## Abstract

Learning representations that remain robust across centuries of variation in handwriting is a key challenge in diachronic representation learning of ancient Greek manuscripts. We introduce three datasets of ancient Greek handwriting for diachronic representation learning: **Hell-Char**, a curated training set spanning the 3rd–1st centuries BCE, and two evaluation sets, **PaLit-Char** (1st–5th c. CE) and **Med-Char** (9th–14th c. CE). To address challenges of symbolic variation, scarce data, and systematic degradation, we propose two methodological innovations: a *similarity-weighted supervised contrastive loss* that biases embeddings by human-perceived confusability, and a *lacuna-driven augmentation* scheme that simulates realistic manuscript corruptions. Trained with these strategies, both a lightweight CNN and a pretrained ResNet achieve strong recognition performance and produce embeddings that more coherently separate character classes than PCA or generic pretrained models. These embeddings enable clustering, identification of stylistic subgroups, and construction of prototype images that visualize diachronic evolution and transitional letterforms. Our results demonstrate that incorporating expert priors and domain-specific corruptions yields robust, interpretable representations, offering a transferable paradigm for representation learning under scarce, temporally evolving, and noisy conditions.

## 1 Introduction

Palaeographic analysis of historical scripts needs strong automated character representation, a problem that remains challenging for scripts such as ancient Greek. Greek handwriting spans over two and a half millennia, encompassing formal literary hands and highly cursive scripts, with substantial variation in stroke shape, scale, slant, and contextual noise (Cavallo, 2009; Crisci & Degni, 2011; Irigoin, 1990; Bianconi, 2015). Material degradation and heterogeneous digitization practices further compound these challenges, introducing ambiguities that complicate segmentation, feature extraction, and character recognition and classification, especially under limited and imbalanced datasets. Although a low-level task, automated character representation has high impact for broader palaeographic analysis, supporting text-image alignment, semi-automatic transcription, and tasks such as script typology, dating, and scribal attribution.

This study addresses this challenge by focusing on the diachronic evolution of ancient Greek letters. We design a lightweight Convolutional Neural Network (CNN) trained with two innovations: a *lacuna-driven augmentation* that simulates realistic manuscript degradations, and a *similarity-weighted supervised contrastive loss* that biases embeddings according to dynamically learned confusability between characters. We evaluate the CNN both in terms of recognition performance and embedding quality. Using confusion matrices, we identify consistently easy or difficult letters and highlight cases of visual confusion. Beyond recognition, clustering analyses on the learned embeddings reveal multiple stylistic subgroups for certain letters, while prototype visualizations per letter–century allow us to study diachronic evolution quantitatively and interpretably. Compared to raw pixels, PCA, or pre-trained features, our CNN embeddings produce a more coherent and discriminative representation of historical Greek handwriting.

We summarize our contributions as four key points:

1. **Historical Greek handwriting datasets:** Three curated datasets spanning the 3rd–14th centuries CE: **Hell-Char** (3rd–1st BCE) for training and benchmarking low-resource, temporally evolving character recognition, and **PaLit-Char** (1st–5th CE) and **Med-Char** (9th–14th CE) for evaluation of generalization across temporal shifts.

2. **Similarity-weighted supervised contrastive loss:** A representation learning objective that biases embeddings according to dynamically learned visual confusability, improving discriminative power for letters with overlapping features.

3. **Lacuna-driven augmentation:** A domain-informed augmentation scheme that faithfully simulates manuscript degradations (lacunae), increasing robustness to missing or corrupted strokes.

4. **Computational paleographic analyses:** Using CNN-derived embeddings, we perform clustering, silhouette-based subgroup detection, and prototype visualization per letter–century, providing interpretable insights into diachronic variation and scribal conventions.

## 2 RELATED WORK

We are not aware of any other study in the literature that analyses the diachronic evolution of Greek handwritten letters between Antiquity and pre-modern times with machine learning. However, we acknowledge the existence of related fields, such as optical character recognition (OCR), and of other investigations on Greek papyri at the character level, which we discuss next.

**OCR**   Early OCR approaches relied on manual feature extraction methods, such as zoning, projection histograms, and contour profiling, to distinguish between characters. A comprehensive survey by Trier et al. (1996) emphasised the importance of these handcrafted features in OCR, while He et al. (2016) introduced a grapheme-based feature extraction system that modelled diachronic variations while incorporating textual features. The advent of deep learning has further transformed the field. LeCun et al. (1998) demonstrated the effectiveness of convolutional neural networks (CNNs) in classifying handwritten digits, laying the groundwork for modern neural approaches in character recognition. Autoencoders (Hinton & Salakhutdinov, 2006) and contrastive learning (Chen et al., 2020) have gained traction in unsupervised learning, enabling models to learn meaningful representations of handwriting directly from data, without the need for manual feature engineering. Leaning on these advances, several of the latter works have examined deep learning for feature analysis of some aspect of ancient Greek handwritings. Marthot-Santaniello et al. (2023) addressed the issue of clustering historical handwriting by similarity with no metadata explicitly indicating date or style. Their method strongly focuses on character-level, employing a SimSiam deep neural network to quantify similarity between images of single Greek letters (Alpha, Epsilon, and Mu) from different manuscripts. Their stylistic similarity observations were useful to palaeographers as they situated manuscripts in an integrated network and disclosed subtle micro-phenomena of similarity.

**CNNs**   Li et al. (2015) applied CNNs to OCR-extracted text, combining visual and textual features to improve dating accuracy. However, their approach assumes the availability of high-quality (historical yet printed) data conducive to accurate OCR results, an assumption that often fails in the context of historical documents such as the Greek papyri addressed in our study. To tackle such challenges, Wahlberg et al. (2016) fine-tuned an ImageNet-pretrained CNN on a corpus of medieval documents, demonstrating improved performance on degraded or irregular scripts. More chronology-specific, West et al. (2024) designed a deep learning pipeline for the automated dating of images of ancient Greek papyrus fragments. Their multi-stage pipeline integrates handwritten text recognition (HTR) for character detection and classification, followed by distinct character-level and fragment-level date prediction models. While single-character dating models are fairly accurate, their aggregated sum of fragment-level models is up to 79% accurate in the prediction of two-century broad date ranges on fragments with large numbers of characters. More recently, Boudraa et al. (2024) proposed a transformer-based pipeline that integrates classical preprocessing techniques with a fine-tuned Vision Transformer and majority-voting for document dating. This study pioneers the integration of Vision Transformers in the context of historical manuscript dating, a domain where CNNs were dominating.

**SimCLR**  Chen et al. (2020) introduced a simple yet powerful contrastive framework for representation learning. Each image is augmented twice, and the network is trained to maximize agreement between positive pairs while treating all other samples in the batch as negatives. While effective as a simple self-supervised technique at scale, SimCLR assumes that all non-matching samples are equally dissimilar. In fine-grained recognition tasks such as character classification, this uniform treatment forces visually similar but distinct classes apart (e.g., A vs. Λ), discarding useful structural information.

**Supervised Contrastive Learning (SCL)**  Khosla et al. (2020) extended SimCLR to the labelled setting by grouping all samples of the same class as positives. This produces tighter class-specific clusters. Importantly, they also showed that combining supervised contrastive embeddings with a linear classifier trained under cross-entropy further improves classification accuracy compared to cross-entropy alone. However, SCL still treats all negatives uniformly, regardless of their visual similarity to the anchor. As a result, classes with inherent affinities (e.g., letters with similar shapes) are repelled too strongly, yielding embeddings that fail to reflect natural inter-class relationships. In addition to instance discrimination, weakly SCL (Zheng et al., 2021) introduced a supervised contrastive component based on weak labels derived from K-nearest neighbor graphs. Instead of treating all other samples as negatives, this approach dynamically identifies semantically similar neighbors and reweights them as positives, alleviating the class collision problem. SCL, treats negatives uniformly and makes classes with inherent affinities (e.g., letters with similar shapes) to be strongly repelled. This leads to embeddings that fail to reflect natural inter-class relationships. This study addresses this gap.

## 3 METHODOLOGY

We analyse handwritten Greek letters from various centuries using CNN-based embeddings trained with SCL enhanced with letter similarity weighting.

### 3.1 CNN BACKBONE

Pavlopoulos et al. (2024) suggested a 2D CNN (fCNN) for dating images of papyri lines, which comprised a fragmentation-based augmentation strategy. We follow a similar fragmentation-based strategy, yet our CNN is different in two ways. First, it is adjusted to operate on letters instead of text lines. Second, the fragmentation augmentation is improved so that synthetic lacunae follow their natural (curvy) shape, i.e., circular or elliptic, not square. The trained model produces high-dimensional embeddings $\mathbf{e} \in \mathbb{R}^D$ representing the visual structure of each letter. The base CNN architecture consists of convolutional layers to extract local stroke and shape patterns; ReLU activations for non-linearity; pooling layers to reduce spatial dimensions while preserving salient features; fully connected layers to map feature maps into the final embedding vector. These embeddings abstract style variations while preserving essential letterform characteristics. We also experiment with ResNet18 pre-trained CNN (He et al., 2016), the ConvNext-V2 self-supervised and globally-normalised CNN Woo et al. (2023), and the ViT-S16 Vision Transformer Caron et al. (2021).

### 3.2 AUGMENTATION

Each character image is converted to grayscale, normalized, and resized to $64 \times 64$ pixels. To account for variability in handwriting and material degradation, we applied rotation (up to $10°$), translation, resizing, color jittering, and lacunae-inspired masking. The lacunae augmentation simulates missing ink or manuscript damage, improving the model's robustness to partial character visibility.

### 3.3 SIMILARITY-WEIGHTED SUPERVISED CONTRASTIVE LOSS

In addition to the standard cross-entropy loss (i.e., the supervised letter-classification objective applied to the backbone's classification head), we train the backbone models using a supervised contrastive loss (SCL), which encourages embeddings of the same letter to cluster together while push-

ing apart visually dissimilar letters. Visual similarities between letters, dynamically learned,[1] are used to weight negative pairs, enabling the model to respect intrinsic inter-letter relationships. This contrastive loss is not computed on the classification logits, but it is applied to the intermediate feature embeddings produced by the backbone before the classification layer. Thus, the model jointly optimises cross-entropy on the classification head and contrastive loss on the shared backbone representations. For each anchor embedding $\mathbf{e}_i$, the loss is defined as:

$$\mathcal{L}_i = -\frac{1}{|P(i)|} \sum_{p \in P(i)} \log \frac{\exp(\mathbf{e}_i \cdot \mathbf{e}_p / \tau)}{\sum_{a \neq i} w_{ia} \exp(\mathbf{e}_i \cdot \mathbf{e}_a / \tau)}$$

where $P(i)$ is the set of positive samples (same class as $i$) and $\tau$ is the softmax temperature; $w_{ia} = 1 + \lambda \frac{S_{y_i, y_a}}{\bar{S}}$ is the weight for negative pair $(i, a)$; $S_{y_i, y_a}$ is the similarity between classes $y_i$ and $y_a$, dynamically computed from embeddings; $\bar{S}$ is the mean off-diagonal similarity; and $\lambda$ controls the influence of similarity weighting. This loss ensures that embeddings of the same letter cluster tightly, while visually similar letters exert weaker repulsion.

### 3.4 PROTOTYPE SELECTION (MEDOID)

For each group (letter, century), we select a representative *medoid* embedding to serve as a prototype ($T$), defined as: $T = \arg\min_i \sum_{j=1}^{N} \left(1 - \cos(\mathbf{e}_c, \mathbf{e}_j)\right)$, where $N$ is the number of embeddings in the group and $e_c$ is the centroid. The medoid ensures a really representative image robust to outliers.

## 4 DATASET DEVELOPMENT

### 4.1 SOURCE

The Hell-Date dataset (Ferretti et al., 2025) comprises 194 images sourced from 157 papyri, all written in Greek and dated between the years 310 BCE and 3 BCE. The material is particularly relevant for digital palaeography and papyrological analysis due to its historical span, script diversity, and accompanying metadata. Each document in the dataset is associated with rich contextual metadata, including the date of composition, the geographical provenance, and the textual type. Of the 194 available images, 171 are annotated at the character level, forming the primary subset of character images used in this work. We used this character-level subset but filtered and restructured it for our purposes; we refer to the restructured subset as Hell-Char. This is the first study to utilize the character annotations included in Hell-Date, which are further presented below.

**The character annotations in Hell-Date** Twenty-nine character classes are present in the annotations of Hell-Date. In addition to the 24 standard letters of the Greek alphabet, the dataset comprises 3 archaic numeral letters (*stigma*, *qoppa*, and *sampi*). It also uses a general 'symbol' category for all characters that are not alphabetic letters. Last, an 'unknown' class was added for uncertain or ambiguous signs, but it remained empty. Each character instance is also assigned a base-type (BT) tag, ranging from BT1 to BT5, which indicates its degree of preservation. These tags can be useful for analysing the correlation between physical degradation and classification performance.

### 4.2 THE HELL-CHAR SUBSET

To reduce the imbalance in character frequency and to ensure a more uniform distribution of samples across classes, we constructed a subset of Hell-Date annotations that we called Hell-Char. Specifically, for each papyrus, at most five instances per character class were randomly selected. The classes for archaic numerals, symbols, and unknown letters were merged into a single general non-alphabetic category ('other'). We limited our analysis to letters tagged BT1 and BT2, which allows excluding characters that are too degraded and are not recognizable out of context. This procedure reduces the dominance of overly frequent letters and mitigates sampling bias across documents.

---

[1]The visual similarities could also be defined manually. Our experiments, however, using a prior similarity matrix based on modern letter shapes, did not lead to improvements.

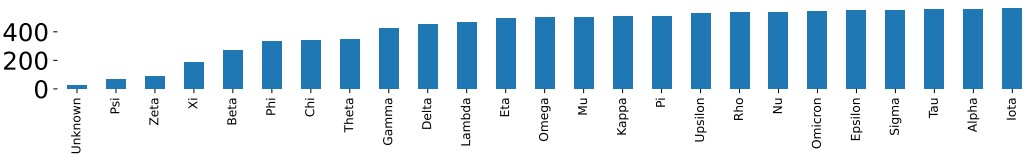

Figure 1: Letter frequency in our Hell-Char subset.

The resulting subset comprises 13,046 character images from 157 distinct papyri. Figure 1 shows the letter frequency in Hell-Char.

**Intrinsic challenges** The dataset presents several intrinsic challenges. The character class distribution is still unbalanced (Figure 1); ambiguous or borderline glyphs can make even human classification difficult. For these characteristics, Hell-Char is a valuable and non-trivial benchmark for evaluating character recognition methods in ancient scripts.

## 5 EMPIRICAL ANALYSIS

### 5.1 EXPERIMENTAL SETTINGS

**The Similarity Matrix** We re-estimate the class-similarity matrix periodically (every 3 train epochs). At each update, we pass the entire training set through the current model, compute class prototypes from the normalized embeddings, and derive the cosine similarity between prototypes (diagonal entries are set to zero). This yields a dynamic measure of inter-class similarity that evolves with the representation space. (An exponential moving average can be applied to stabilize updates.) The updated matrix is then used by our Dynamically Supervised Contrastive Loss (DSCL), which down-weights negatives from highly similar classes and up-weights negatives from dissimilar ones.

**Lacuna-driven Synthetic Fragmentation** We attempt to simulate manuscript deterioration more realistically than standard erasure augmentations by inserting irregular regions that approximate actual lacunae observed in historical documents. For each image, we sample 1–4 lacunae and each covers 2–15% of the area to match the typical size distribution of physical papyrus damage. Lacuna geometry is obtained by drawing anisotropic ellipses whose contours are further distorted via random morphological operations (erosion or dilation), producing organic, non-rectangular shapes characteristic of flaking, humidity damage, parchment wear, or insect deterioration (e.g., worm holes are frequent in papyri). These lacunae are placed at random positions and the masked pixels are replaced with background values, reflecting the absence of ink/papyrus rather than additive noise. This augmentation increases robustness to fragmentary handwriting and introduces realistic variability at negligible computational cost.

**Data Split** We keep 20% of the data for testing, following a letter-based stratified split. Although we acknowledge that this strategy allows a scribe-based leakage, the selected approach fits better the scope of this work (see Appendix A.4).

### 5.2 LETTER RECOGNITION

Table 1 shows the performance of backbones when we add fragmentation-based augmentation and contrastive loss. A vanilla CNN, as in Pavlopoulos et al. (2024) but without any fragmentation, achieves an Accuracy of 74%. F1 is exactly the same, indicating the balanced performance across letters despite the class imbalance (Figure 1). The model of Pavlopoulos et al. (2024) performs better than the same model with random erasure in both metrics, but our Lacunae-based augmentation outperforms both. The architecture of ResNet18 (He et al., 2016), when trained from scratch, performs worse in F1 and on par in Accuracy. Pre-trained, however, it outperforms all the models above. When we enhance fCNN with our LF and dynamically-weighted supervised contrastive loss, it outperforms the pre-trained ResNet18. But when we enhance the latter with our Lacunae-based augmentation and our similarity contrastive loss, we achieve the best results. Per-letter classification

Table 1: Classification performance on Hell-Char (sorted) of fCNN (Pavlopoulos et al., 2024) and ResNet18 (He et al., 2016), pre-trained (PT) and/or fine-tuned (FT), when we add: our SCL with dynamically-learned weights, and fragmentation-based augmentation (none, random, our LF).

| Model | Fragmentation | Contrastive Loss | **Accuracy** | **F1** |
|---|---|---|---|---|
| fCNN | - | - | 0.742 | 0.74 |
| fCNN | Random | - | 0.768 | 0.75 |
| fCNN | LF | - | 0.782 | 0.77 |
| ResNet18-FT | - | - | 0.788 | 0.74 |
| ResNet18-PT+FT | - | - | 0.801 | 0.79 |
| fCNN | LF | DSCL | 0.803 | 0.80 |
| ResNet18-PT+FT | LF | DSCL | **0.829** | 0.82 |

performance is provided in Appendix A. LF and DSCL improve also the classification performance of ViT-16S and ConvNeXt-V2 (Appendix E.1), yet they overfit and are not further analysed.

## 5.3 LETTER IMAGE CLUSTERING

We observe that CNN image embeddings can be used to represent letters. To assess the quality of the resulting embeddings, we compared them against baseline features, then feeding algorithms that should cluster images of the same letter into subcategories. We also engineered features, based on Otsu's method (Otsu, 1979), a widely used adaptive thresholding technique, and principal component analysis (PCA) (Karl, 1901), keeping as many dimensions as add up to 90% of the original information (i.e., 500). Characters have consistent alignment and size, hence pixel-based variance captured by PCA can correspond to meaningful features of the characters (e.g., strokes and overall shape). Although it destroys 2D structure (edges, texture) and does not focus on separability, PCA applied to the raw input is a simple preprocessing baseline that is complementary to CNN features that preserve local structure. The empirical results, shown in Table 2, underscore the importance

Table 2: Clustering performance on Hell-Char using different embeddings and different clustering algorithms, sorted by performance of the best performing Spectral algorithm.

| | k-means | | Spectral | | AH | |
|---|---|---|---|---|---|---|
| Embedding | NMI | ARI | NMI | ARI | NMI | ARI |
| ResNet18+LF+DSCL | **0.667** | **0.411** | **0.836** | **0.743** | **0.818** | **0.726** |
| fCNN+LF+DSCL | 0.428 | 0.189 | 0.631 | 0.442 | 0.544 | 0.292 |
| ResNet18+PT+FT | 0.480 | 0.257 | 0.487 | 0.225 | 0.464 | 0.197 |
| Otsu+PCA | 0.318 | 0.152 | 0.382 | 0.176 | 0.356 | 0.168 |
| ResNet18+PT | 0.067 | 0.010 | 0.094 | 0.015 | 0.073 | 0.008 |

of task-specific embeddings and non-linear clustering for historical handwriting. Our ResNet18, enhanced with our proposed LF and SCL, consistently outperforms both Otsu+PCA and the pre-trained ResNet18, achieving markedly higher agreement with paleographic labels across all metrics. Otsu+PCA, though superior to raw pretrained features, lags far behind, while ResNet18 fails entirely, with near-random partitions. The stark contrast highlights two key findings: (i) general-purpose CNN features trained on modern image corpora do not transfer to paleographic tasks, and (ii) the manifold structure of handwritten letter embeddings is not captured adequately by centroid-based partitioning. Together, these results validate the need for domain-tailored architectures and manifold-aware clustering to recover meaningful structure in diachronic handwriting data.

## 5.4 PATTERN RECOGNITION: REVEALING LETTER FORMS

Using Spectral Clustering on the embeddings of our best performing CNN (ResNet18+LF+DSCL; see Table 1), we applied the Silhouette method (Schubert, 2023) to detect the optimal number of clusters per letter. For each letter, we varied the number of clusters and retained the configuration

with the highest Silhouette score (Rousseeuw, 1987). For one letter (Alpha), the optimal number exceeded the two clusters, indicating multiple distinct forms.[2] The resulting letter forms (cluster medoids) are shown in Figure 2.

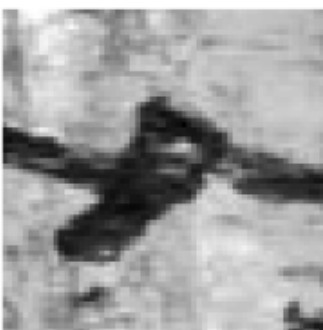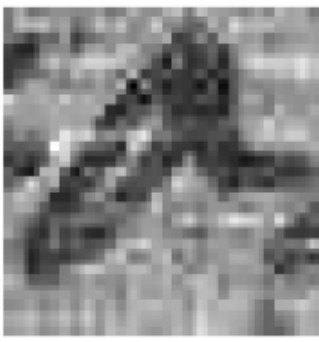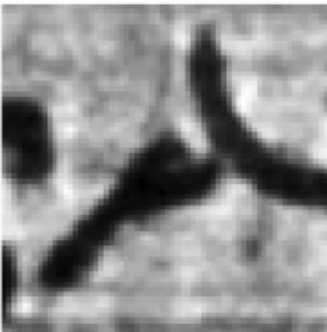

Figure 2: Representative forms of the Greek letter Alpha for which three clusters were detected using the Silhouette method. Forms for other letters are shown in Appendix B.

Clustering letter forms into subtypes is a hard and unsolved task in paleography. The results of the network may in some cases point to useful characteristics. In the case of Alpha, for which we showed that several forms exist, the three images seem to represent different characteristics: the first alpha is filled-in (no empty space in its centre), circular and ligatured to the left; the second one is circular but not ligatured; the third one is angular. This partition is coherent from a paleographical point of view. Examples of the representative forms per letter for all letters are in Appendix B.

## 6 OUT OF TEMPORAL DISTRIBUTION APPLICATION

The backbone CNN models in this work are trained on letter images from papyri of the last three centuries BCE. In this period, the epigraphic letter forms (close to our modern capital letters) start to be modified with increasing cursivity, driven by the practical demands of faster writing. This increase in cursivity continues over the following centuries and constantly deforms letter shapes; however, epigraphic letter forms are maintained, especially for calligraphic writing styles called capital (or uncial) bookhands. In the 9th century, the calligraphic stylisation of cursive forms that had gradually developed over the previous centuries reached the so-called state of "minuscule script". While many minuscule letterforms remain visually close to their capital ancestors, others diverge significantly (notably Beta, Mu, Gamma, and Delta). During the following centuries, uncial and minuscule calligraphic forms continued to coexist, sometimes within the same manuscript and even within a single word.

### 6.1 EVALUATION DATASET DEVELOPMENT

**PaLit-Char: Majuscule Literary Papyri** To evaluate how well the model generalizes to letter-forms close in time to the training data, we constructed the **PaLit-Char** test set. It is a fully balanced dataset containing 384 images (4 specimens × 24 letters × 4 centuries) spanning the 2nd–5th CE. Images were drawn from securely dated literary papyri in the PaLit dataset (Pavlopoulos et al., 2024); for the 5th century, where securely dated material is scarce, 48 images were taken from an additional, palaeographically dated manuscript. While Hell-Char covers cursive handwritings from the last three centuries BCE, PaLit-Char extends into the early centuries CE and covers calligraphic writing, offering both chronological continuity and stylistic diversity. This allows us to test whether features learned on late Hellenistic cursive letters transfer to Roman-period bookhands that retain strong ties to their predecessors but already display variation.

---

[2]Silhouette scores cannot be computed for a single cluster; hence the minimum number considered was two. For the remaining letters, additional sub-forms may still exist.

**Med-Char: Medieval Minuscule Manuscripts**   With the historical evolution described above in mind—and having first tested the recognition performance of our network on the chronologically close PaLit-Char—we proceed to test its ability to recognize letterforms from medieval minuscule manuscripts. This evaluates both the generalizability of learned features across palaeographic periods and the limits of shape-based classification given the diachronic script variation. To assess this hypothesis, we compiled a dataset of 574 letter images from manuscripts dated between 835 and 1378 CE, a much later period. We used 24 images per letter, opting for balance across the centuries in that period,[3] and using the best performing ResNet18, enhanced with our LF and SCL, to classify each image. We call this evaluation dataset **Med-Char**. Contrary to our training set and the PaLit-Char test set, which contain capital or cursive letters (upper case), Med-Char is a Byzantine minuscule letter (lower case) dataset. This choice is deliberate: minuscule script is historically derived from majuscule but exhibits substantial graphic divergence, with some letters retaining visual continuity and others undergoing radical transformation. Testing on Med-Char therefore allows us to probe the limits of the learned representations under extreme diachronic and stylistic shift. This provides a benchmark for cross-period generalization.

## 6.2 EXPERIMENTAL ANALYSIS

**Closer in Time**   On the evaluation data of PaLit-Char, ResNet18+LF+DSCL achieves an Accuracy and F1 of 0.84, very close to the results of Hell-Char. This is reasonable due to the proximity in time and nature (the full classification report is in Table 4 in the Appendix). Although F1 dropped for specific letters (e.g., Phi, Pi, Psi) for others it improved (Alpha and Zeta). The calligraphic nature (regular, standardized, legible) of PaLit characters can explain this increase.

**Far Away in Time**   ResNet18+LF+DSCL achieves an Accuracy of 0.45 in Med-Char, revealing a highly uneven performance across the 24 character classes (see Table 5 in the Appendix). Letters Chi, Epsilon, Iota and Lambda achieve high F1 (0.88, 0.70, 0.73, and 0.84 respectively), indicating that the network captures their discriminative features reliably despite temporal variability. Indeed, for these letters, capital, cursive and minuscule letter shapes are similar to one another. In contrast, other letters (Alpha, Delta, Gamma, Upsilon) exhibit extremely low or even zero F1 values, suggesting systematic confusion with visually similar shapes and high diachronic variability that undermines generalization. Gamma, for instance, undergoes a strong visual evolution; in Hell-Char, its shape is specific and close to epigraphic Γ, whereas in Med-Char, it is very different and rather resembles Upsilon. The remaining letters fall into an intermediate band, with varying degrees of precision–recall trade-offs: e.g., Kappa, Omicron and Tau show strong Recall (0.75, 0.83 and 0.83) but lower Precision (0.39, 0.39 and 0.42), while Psi leads to the highest Precision (1.00) yet inflated Recall (0.14), reflecting over-prediction. As can be seen in Figure 3, misclassification patterns are temporally structured: errors for Chi are closer to 1300 CE, whereas Iota's confusion is around 950 CE, implying that historical morphological shifts exert non-uniform effects on recognition difficulty. Similar patterns occurs for Tau (around 1250 CE) and Theta (1000 CE). Noteworthy is the fact that fine-tuning on Palit-Char and inferring on Med-Char brings no significant gains (see Appendix D).

**Letter-Century Clusters**   Figure 4 illustrates a two-dimensional t-SNE projection of the ResNet18+LF+DSCL embeddings, where each point corresponds to an image patch representing a handwritten Greek Med-Char character. To reduce clutter and improve interpretability, instead of showing all individual samples, one prototype image per letter–century pair is overlaid: the prototype is chosen as the sample closest to the centroid of its group in the t-SNE space, thus representing the most "typical" example of that cluster. The resulting map highlights how temporal and graphemic factors shape the embedding space. Within the clusters related to one character, overlapping or diffuse areas indicate stylistic continuities or transitional forms between centuries, whereas sharp separations reveal periods of stronger diachronic variation. This approach provides an interpretable way of assessing the alignment between automated embeddings and paleographic expectations, enabling both qualitative validation of the clustering behavior and the identification of anomalies or particularly distinctive exemplars.

---

[3]We include 24 random instances of each letter per century from multiple manuscripts. Letter Psi was less supported and has 22 occurrences.

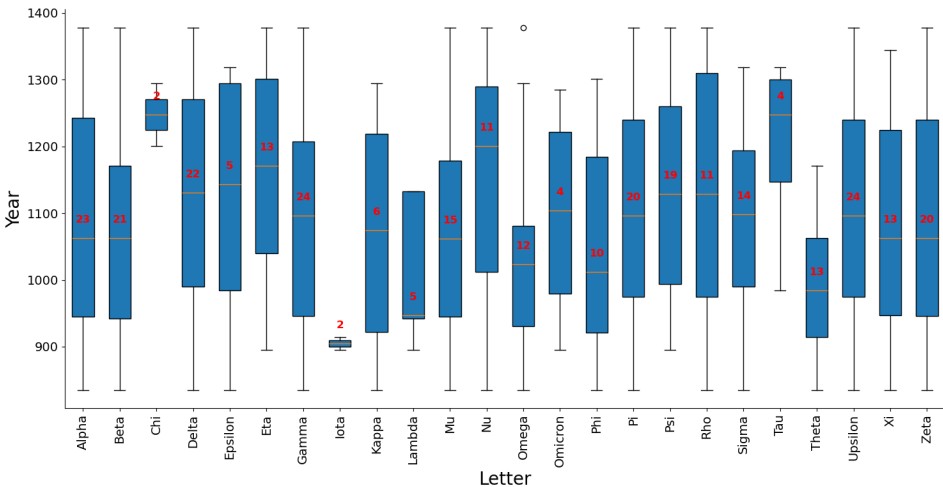

Figure 3: Boxplot of years per letter for missclassified out-of-distribution images of Med-Char. The count of mistakes is shown inside in red letters.

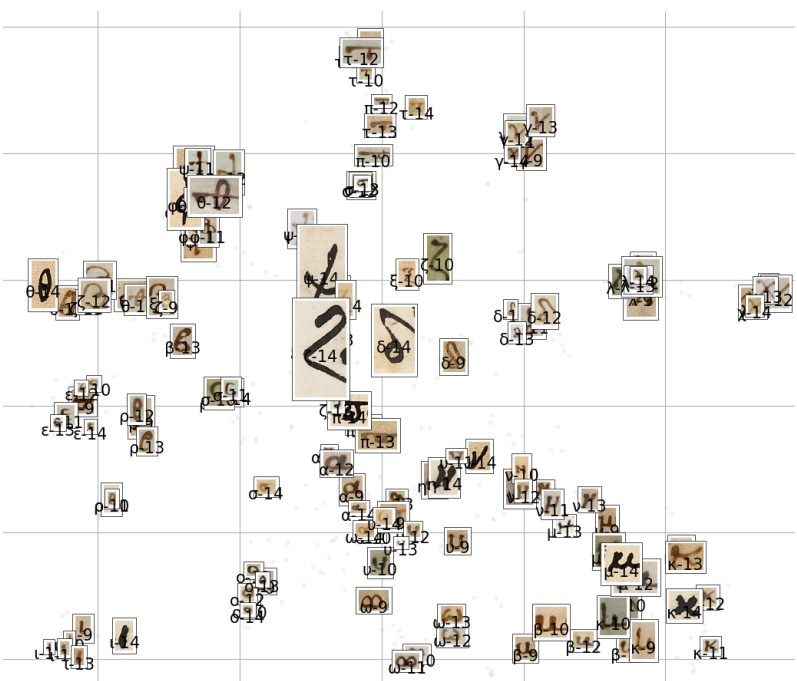

Figure 4: Two-dimensional t-SNE plot of ResNet18+LF+DSCL embeddings on Med-Char. One prototype image per letter-century group is shown, selected as the sample closest to the centroid, to visualize the cluster structure across both graphemic and temporal dimensions.

**Distinctively Isolated Letters** The letters that obtained high F1 scores (i.e., Chi, Epsilon, Iota and Lambda), are distinctively isolated in clusters on the exterior of the graph. Their shapes are close to Hellenistic ones from the training set Hell-Char. Also, Gamma is grouped, but its shape is very different from Hellenistic Hell-Char Gamma shape and closely resembles Hellenistic Hell-Char Upsilon and Tau shapes; this can explain the zero Precision and Recall for Gamma. **Visible Clusters: The cases of Zeta and Beta** Two clusters of Zeta are visible. A first one to the left of the graph, mixed with Theta, with a shape that resembles a 3. A second one in the middle of the graph, mixed with Delta and Xi, with a shape close to modern-day $\zeta$. This distinction points out to one reason for confusion in recognizing Zeta: its "3-looking" shape is absent from the training data and therefore,

cannot be identified. The same is true for Beta: its B-shaped form groups with Zeta to the left of the image, whereas its minuscule, u-shaped form groups with other minuscule u-shaped letters such as Kappa, Mu and Nu to the bottom right of the graph. This u-shaped Beta, absent from Hell-Char, explains its very low Recall. **Typical Medieval Forms** The bottom-right corner of the graph, with worse clustering for individual letters, groups typical Medieval letter forms, based on successions of 'o' and 'u' shapes. These shapes are quite different from Hell-Char letters shapes, and indeed the letters represented here (Omega, Beta, Kappa, Mu, Nu, Upsilon) do not belong to the top-performing ones. Kappa and Nu achieved an F1 of 0.51, which can be explained by their mixing Medieval, minuscule, u-shaped forms with older, capital forms already attested in Hell-Char. These forms, K and N, can be seen at the margins of the larger cluster on the bottom-right corner.

# 7 DISCUSSION

**Novel Methodological Contributions**   Erasing input as augmentation in image classification is not new Zhong et al. (2020) and it has been shown particularly useful for papyri, which are often fragmented. **Our presented synthetic augmentation** is closer in nature to the real fragments (i.e., elliptic v. square) and a comparison between rows 2-3 of Table 7 shows that our approach is better. **Our proposed SW for SCL**, on the other hand, besides interpretability (representation class similarity), helps the model avoid confusion. In Figure 6, for example, only one (alpha-lambda) out of the four pairs of high similarity noted in the caption get a high value in the confusion matrix (Figure 5). Standard SCL treats negatives uniformly, making classes with inherent affinities (e.g., letters with similar shapes, such as as psi-phi) to be strongly repelled. This leads to embeddings that fail to reflect natural inter-class relationships. Our results (Table 7) reflect the superiority of our approach.

**Scribe Leakage**   We frame paleographic problems (e.g., dating or scribe identification) as the search for a strong script embedding, where classification relies on defining distance thresholds (same period or scribe). Our current work validates robust letter representations using classification. Testing against a chronologically distinct external dataset (Table 4) confirmed the absence of validation leakage; i.e., accuracy did not drop. While this validates model integrity, it confirms that our representations do not capture the fidelity required for scribal identification, which remains a challenging task, very hard to solve D'Alessandro et al. (2025). This limitation is likely imposed by the low sample density (i.e., a maximum of 120 characters per papyrus).

**Resources Contribution**   Besides our two technical contributions, we also release Med-Char and PaLit-Char, two new datasets. Hell-Char is a subset of an existing dataset called Hell-Date, already released in the past yet never used for applications to the best of our knowledge. Together, these three resources are expected to assist the field of computational palaeography.

# 8 CONCLUSIONS

This work introduces three datasets of historical Greek handwriting (**Hell-Char**, **PaLit-Char**, **Med-Char**), and uses them to examine how modern representation learning captures symbolic variation across time. Beyond establishing Hell-Char as a benchmark for low-resource, domain-shifted visual recognition, we propose two methodological innovations: **a similarity-weighted supervised contrastive loss**, which aligns representations with human-perceived character confusability, and **a lacuna-driven augmentation scheme**, which faithfully simulates manuscript degradations. Empirically, we show that CNN-derived embeddings yield a more discriminative structure than PCA or generic pre-trained models, while clustering uncovers stylistic subgroups that mirror diachronic variation and coexisting scribal conventions. Prototype distribution per letter–century further visualize gradual graphical change, providing interpretable bridges between computational analysis and paleographic interpretation. More broadly, this study highlights the limits of natural-image transfer learning for specialized domains and demonstrates how integrating expert prior knowledge with domain-specific corruptions can produce robust and faithful embeddings. The resulting framework is not only valuable for computational paleography but also constitutes a transferable paradigm for **representation learning under scarce, temporally evolving, and systematically corrupted data**.

## REPRODUCIBILITY STATEMENT

Our code and data will be released publicly (CC license) upon acceptance. An anonymized GitHub repository is made for reviewing purposes at https://anonymous.4open.science/r/letter-evol/ including the source code (`source.py`), data samples (cliplets and CSV per dataset), and notebooks with training and evaluation pipelines.

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

# A  PER LETTER CLASSIFICATION

## A.1  PERFORMANCE

Table 6 shows the classification performance per letter of the best performing ResNet18, pre-trained and enhanced with our supervised contrastive loss (SCL) with dynamically learned weights and

our Lacunae-based fragmentation (LF). The model achieves 0.83 in Accuracy, with macro- and weighted-F1 following closely, indicating the balanced performance across the 24 Greek letters despite the class imbalance. Several characters are classified with very high F1, e.g. Beta (0.92), Eta (0.92), Kappa (0.92), Omicron (0.90), Chi (0.89), Nu (0.89), Rho (0.89). Letters such as Alpha (0.63 F1), Lambda (0.63 F1) and Zeta (0.70 F1) underperform. Low support may explain why Zeta underperforms (22 instances). Mid-performing classes, such as Theta (0.73 precision, 0.79 recall, 0.76 F1), indicate a possible difficulty of capturing internal script characteristics. Among the circularly-shaped letters (Theta, Omicron, Epsilon and Sigma), Theta is the rarest and is often influenced by the shapes of the others, thus creating sources for confusion.

Table 3: The classification report on Hell-Char, per letter, of ResNet18 enhanced with our LF augmentation and our SCL with dynamically-learned weights.

| Class | Precision | Recall | F1-Score | Support |
|---|---|---|---|---|
| Alpha | 0.73 | 0.55 | 0.63 | 139 |
| Beta | 0.92 | 0.91 | 0.92 | 67 |
| Chi | 0.97 | 0.82 | 0.89 | 85 |
| Delta | 0.87 | 0.84 | 0.85 | 113 |
| Epsilon | 0.84 | 0.89 | 0.86 | 134 |
| Eta | 0.92 | 0.91 | 0.92 | 128 |
| Gamma | 0.83 | 0.75 | 0.79 | 105 |
| Iota | 0.89 | 0.72 | 0.79 | 141 |
| Kappa | 0.89 | 0.94 | 0.92 | 127 |
| Lambda | 0.63 | 0.64 | 0.63 | 136 |
| Mu | 0.83 | 0.87 | 0.85 | 126 |
| Nu | 0.83 | 0.97 | 0.89 | 134 |
| Omega | 0.84 | 0.82 | 0.83 | 123 |
| Omicron | 0.88 | 0.92 | 0.90 | 126 |
| Phi | 0.88 | 0.84 | 0.86 | 83 |
| Pi | 0.80 | 0.91 | 0.85 | 127 |
| Psi | 0.83 | 0.89 | 0.86 | 17 |
| Rho | 0.86 | 0.91 | 0.89 | 133 |
| Sigma | 0.74 | 0.90 | 0.81 | 138 |
| Tau | 0.73 | 0.85 | 0.79 | 139 |
| Theta | 0.73 | 0.79 | 0.76 | 86 |
| Upsilon | 0.82 | 0.73 | 0.77 | 133 |
| Xi | 0.76 | 0.83 | 0.80 | 47 |
| Zeta | 0.78 | 0.64 | 0.70 | 22 |
| Accuracy | | | 0.83 | 2603 |
| Macro (avg) | 0.84 | 0.82 | 0.82 | 2603 |
| Weighted (avg) | 0.83 | 0.83 | 0.83 | 2603 |

## A.2 CONFUSION

As it is apparent in the confusion matrix (Fig. 5), images of Alpha and Lambda were hard to classify, possibly due to their visual similarity. Alpha VS Delta, however, as well as Lambda VS Delta, which are also similar, are not confused. Other confused pairs are Iota vs Rho, Sigma VS Epsilon (but not Epsilon VS Sigma), Theta VS Omicron (but not Omicron VS Theta), Upsilon VS Tau, Xi VS Zeta, all pairs with strong visual similarities that are indeed dynamically assigned a high similarity during training. The dynamically learnt similarity matrix is provided in Figure 6. The Sigma-Epsilon confusion can explain why Sigma has a very high recall (0.90) but a low precision (0.74, for an F1 of 0.81).

## A.3 COMPENSATION

Irigoin (1990, p. 303-304) proposed that ancient readers compensated for the confusion between a consonant and a vowel through their language knowledge, so that a graphic system needed to

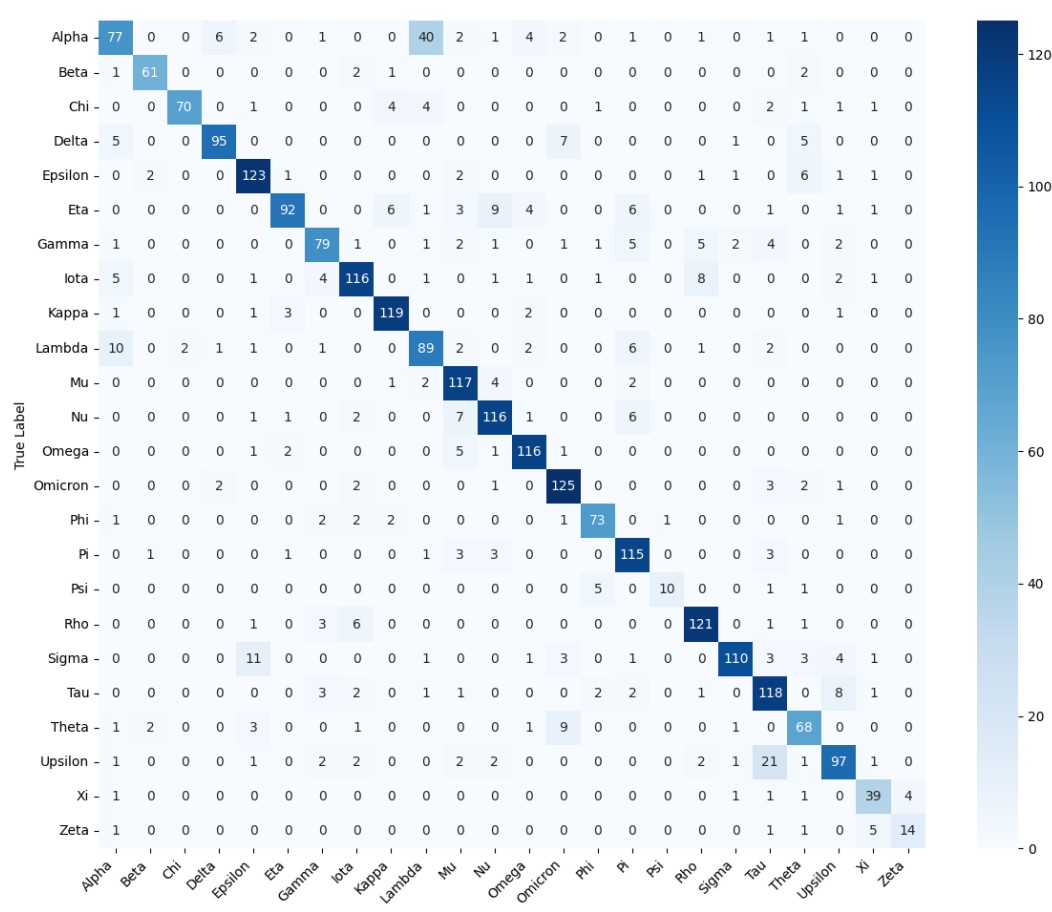

Figure 5: Confusion matrix of ResNet18 with LF and SCL.

maximize the difference between letters with phonologically similar functions (vowels between themselves, or consonants between themselves). The confusion matrix (Figure 5) confirms this hypothesis; thus, Lambda (a consonant) is confused with Alpha (a vowel) but less with Delta (another consonant); similarly, Theta (a consonant) is confused with Omicron (a vowel) but less with Sigma (a consonant). Therefore, the results in clustering show that our method represents letters in a way that is paleographically significant and can help paleographers navigate questions of readability of a script based on confusion patterns.

### A.4 OUT OF DISTRIBUTION

The performance of ResNet18+LF+DSCL on out of distribution datasets is shown in Tables 4-5. In PaLit-Char, Accuracy is high across letters except from Psi, which is confused with Phi. In the much later in time Med-Char, Accuracy drops to 0.45 and Psi drops further, along with several other letters. Exceptions are Chi, Epsilon, Iota, Lambda.

## B LETTER FORM RECOGNITION

Figure 7 shows the two representative forms per letter for the (23) letters besides Alpha. Given that the Silhouette method operates for more than two clusters, we observe that either one or two letter forms exist per letter.

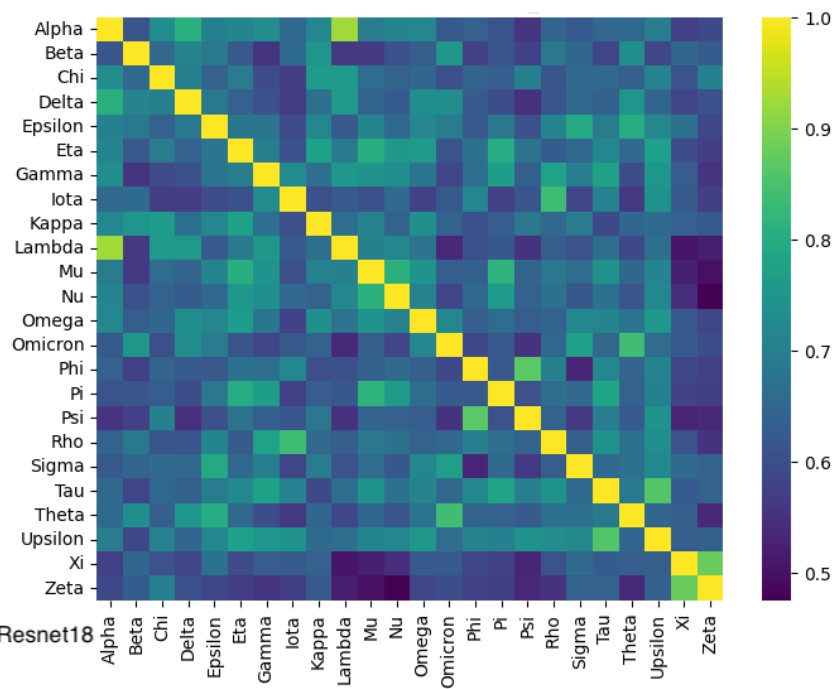

Figure 6: Dynamic similarity matrix between letters learned during training. Light colours indicate high similarity, such as Alpha-Lambda, Theta-Omicron, Xi-Zeta, Phi-Psi.

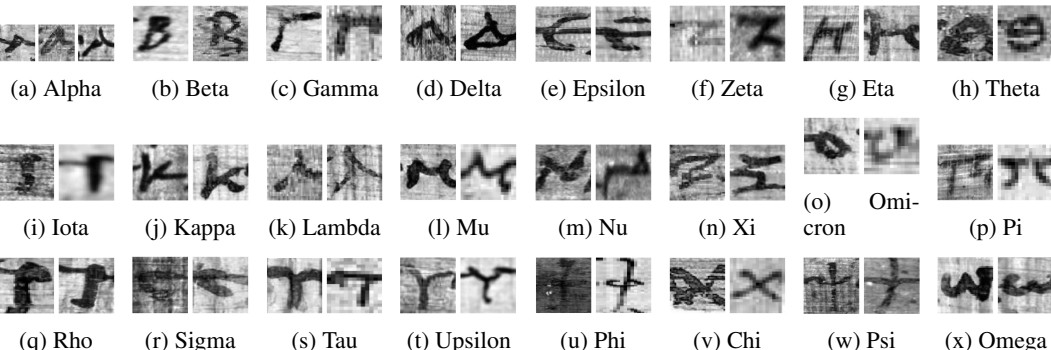

| (a) Alpha | (b) Beta | (c) Gamma | (d) Delta | (e) Epsilon | (f) Zeta | (g) Eta | (h) Theta |

| (i) Iota | (j) Kappa | (k) Lambda | (l) Mu | (m) Nu | (n) Xi | (o) Omi-cron | (p) Pi |

| (q) Rho | (r) Sigma | (s) Tau | (t) Upsilon | (u) Phi | (v) Chi | (w) Psi | (x) Omega |

Figure 7: Representative forms per Greek letter. Only in Alpha three forms were found. For each other letter, the shown forms may or may not indicate distinct subforms.

## C    EXPERIMENTAL SETUP

In subsection 5.2, we presented the classification performance of different configurations.

### C.1    MODELS

The fCNN architecture comprises three convolutional blocks with 32, 64, and 128 channels, respectively, each employing $3 \times 3$ kernels, ReLU activation functions, and $2 \times 2$ max-pooling. These are followed by a fully connected layer with 512 hidden units, dropout (p = 0.5), and a final softmax classification layer. For the ResNet18 baseline, we adopt the standard architecture proposed by He et al. (2016), initialized with ImageNet-pretrained weights. The Swin Transformer is adapted by modifying its initial convolutional layer to accommodate single-channel grayscale inputs and by re-

Table 4: Classification performance of ResNet18+LF+DSCL on PaLit-Char

| Class | Precision | Recall | F1-Score | Support |
|---|---|---|---|---|
| Alpha | 0.58 | 0.94 | 0.71 | 16 |
| Beta | 0.94 | 1.00 | 0.97 | 16 |
| Chi | 1.00 | 0.88 | 0.93 | 16 |
| Delta | 1.00 | 0.75 | 0.86 | 16 |
| Epsilon | 0.88 | 0.88 | 0.88 | 16 |
| Eta | 0.93 | 0.88 | 0.90 | 16 |
| Gamma | 0.87 | 0.81 | 0.84 | 16 |
| Iota | 0.89 | 1.00 | 0.94 | 16 |
| Kappa | 0.88 | 0.94 | 0.91 | 16 |
| Lambda | 0.71 | 0.67 | 0.69 | 15 |
| Mu | 0.94 | 0.94 | 0.94 | 16 |
| Nu | 0.75 | 0.94 | 0.83 | 16 |
| Omega | 1.00 | 0.94 | 0.97 | 16 |
| Omicron | 0.93 | 0.88 | 0.90 | 16 |
| Phi | 0.64 | 0.88 | 0.74 | 16 |
| Pi | 0.93 | 0.81 | 0.87 | 16 |
| Psi | 1.00 | 0.19 | 0.32 | 16 |
| Rho | 0.74 | 0.88 | 0.80 | 16 |
| Sigma | 0.87 | 0.81 | 0.84 | 16 |
| Tau | 0.65 | 0.69 | 0.67 | 16 |
| Theta | 0.79 | 0.94 | 0.86 | 16 |
| Upsilon | 0.88 | 0.88 | 0.88 | 16 |
| Xi | 0.88 | 0.94 | 0.91 | 16 |
| Zeta | 1.00 | 0.75 | 0.86 | 16 |
| Accuracy | 0.84 | 0.84 | 0.84 | 383 |
| Macro (avg) | 0.86 | 0.84 | 0.83 | 383 |
| Weighted (avg) | 0.86 | 0.84 | 0.83 | 383 |

placing the default classification head with a custom layer designed to output predictions across 24 target classes.

## C.2 TRAINING

We used the Adam optimizer with default parameters ($\beta_1 = 0.9$, $\beta_2 = 0.999$). The learning rate was set to 0.001 for the fCNN and 0.0001 for the pre-trained ResNet and SWIN, to avoid catastrophic forgetting. All experiments were conducted with a batch size of 16 and trained for up to 100 epochs, with early stopping applied if the validation loss did not decrease for 10 consecutive epochs. Additionally, we employed a ReduceLROnPlateau scheduler to adjust the learning rate during training.

## D CROSS-DATASET FINE-TUNING

The results presented in Table 9 show that ResNet18+LF+DSCL, trained on Hell-char and fine-tuned on PaLit-char, underperforms on Med-Char. This phenomenon can be explained by understanding the evolution of Greek handwriting over the millennia. Indeed, our three datasets Hell-Char, PaLit-Char and Med-Char differ not only based on their chronology, but also based on their formal characteristics.

In Classical Greece (ca. $5th - 4th$ century BCE), the Greek alphabet had clear, separated letter shapes that resemble what we call today upper-case or majuscule. Thus, Gamma would look like $\Gamma$, and Delta would look like $\Delta$. These shapes were easy to read but slow to write.

During the Hellenistic period (ca. $3rd - 1st$ century BCE), these slow and clear forms were retained for calligraphic book writing. However, everyday writing with ink developed rapidly written cursive forms of the majuscule that progressively diverged from earlier shapes. These cursive, unstable shapes form the backbone of our training set, Hell-Char. They resemble nothing used in modern-day typography; some examples can be seen in Figure 7.

Table 5: Classification performance of ResNet18+LF+DSCL on Med-Char.

| Class | Precision | Recall | F1-Score | Support |
|-------|-----------|--------|----------|---------|
| Alpha | 0.04 | 0.04 | 0.04 | 24 |
| Beta | 0.50 | 0.12 | 0.20 | 24 |
| Chi | 0.85 | 0.92 | 0.88 | 24 |
| Delta | 0.20 | 0.08 | 0.12 | 24 |
| Epsilon | 0.63 | 0.79 | 0.70 | 24 |
| Eta | 0.61 | 0.46 | 0.52 | 24 |
| Gamma | 0.00 | 0.00 | 0.00 | 24 |
| Iota | 0.61 | 0.92 | 0.73 | 24 |
| Kappa | 0.39 | 0.75 | 0.51 | 24 |
| Lambda | 0.90 | 0.79 | 0.84 | 24 |
| Mu | 0.60 | 0.38 | 0.46 | 24 |
| Nu | 0.48 | 0.54 | 0.51 | 24 |
| Omega | 0.29 | 0.50 | 0.36 | 24 |
| Omicron | 0.39 | 0.83 | 0.53 | 24 |
| Phi | 0.44 | 0.58 | 0.50 | 24 |
| Pi | 0.36 | 0.17 | 0.23 | 24 |
| Psi | 1.00 | 0.14 | 0.24 | 22 |
| Rho | 0.68 | 0.54 | 0.60 | 24 |
| Sigma | 0.40 | 0.42 | 0.41 | 24 |
| Tau | 0.42 | 0.83 | 0.56 | 24 |
| Theta | 0.32 | 0.46 | 0.38 | 24 |
| Upsilon | 0.00 | 0.00 | 0.00 | 24 |
| Xi | 0.73 | 0.46 | 0.56 | 24 |
| Zeta | 0.67 | 0.17 | 0.27 | 24 |
| Accuracy | | | 0.45 | 574 |
| Macro (avg) | 0.48 | 0.45 | 0.42 | 574 |
| Weighted (avg) | 0.48 | 0.45 | 0.42 | 574 |

In the Roman and Late Antique period (ca. $1\text{st} - 8\text{th}$ century CE), calligraphic books continued to use slow, clear uppercase letter forms attested since the Classical times, with small stylistic variations (slant, proportions, thick and thin strokes...). These are the letter shapes present in PaLit-Char, our fine-tuning set. However, cursive handwriting continued to evolve, drastically changing letter shapes over time. These changes are undocumented in our datasets and thus escaped both training and fine-tuning.

Around the $9\text{th}$ century CE, a later formal script, the so-called minuscule script, was developed when the appearance of the then-contemporary fast documentary hand was standardized and adopted for book production (Cavallo, 2009, p. 136). This script is based on the later cursive shapes that are undocumented in our training and fine-tuning sets; it is close to modern-day lowercase Greek, with Gamma looking like $\gamma$ and Delta like $\delta$. Some of its letter shapes are still visually similar to majuscule forms (e.g. Omicron, $O$ and $o$), but others differ drastically (e.g. $\Gamma$ and $\gamma$). During the $9\text{th} - 14\text{th}$ century CE, this script coexisted with majuscule shapes even within the same manuscript. This mix of minuscule with some majuscule contamination is the script of our test set Med-Char.

These massive, historically induced changes in character shapes can explain why, even with fine-tuning, the model failed to generalise across time. Specifically, a model trained on Hell-Char (an early fast cursive) and subsequently fine-tuned on PaLit-Char (a formal majuscule) achieves proficiency in both calligraphic majuscule forms (e.g., $A, B, \Gamma$) and the complex Hellenistic cursive shapes. However, when this resulting model is tasked with classifying Med-Char (a formal minuscule, e.g., $\alpha, \beta, \gamma$), it struggles to generalize the morphological shift, as it lacks exposure to the intermediary Roman and Late Antique cursive scripts—the evolutionary foundation from which minuscule scripts were later formalized.

This can lead to systematic misclassification, as the model attempts to find the closest known visual proxy instead of leveraging the absent evolutionary steps of later cursive shapes. For instance, Gamma (Precision, Recall and F1 of 0.00) was presumably confused with Upsilon, as Med-Char Gamma (similar to $\gamma$) is very different from Hell-Char and PaLit-Char Gamma (similar to $\Gamma$) and

Table 6: Classification performance of ViT+LF+DSCL on Med-Char.

| Class | Precision | Recall | F1-Score | Support |
|---|---|---|---|---|
| Alpha | 0.00 | 0.00 | 0.00 | 24 |
| Beta | 0.03 | 0.04 | 0.04 | 24 |
| Chi | 0.53 | 0.83 | 0.65 | 24 |
| Delta | 0.50 | 0.04 | 0.08 | 24 |
| Epsilon | 0.42 | 0.46 | 0.44 | 24 |
| Eta | 0.47 | 0.33 | 0.39 | 24 |
| Gamma | 0.00 | 0.00 | 0.00 | 24 |
| Iota | 0.37 | 0.75 | 0.49 | 24 |
| Kappa | 0.28 | 0.83 | 0.42 | 24 |
| Lambda | 1.00 | 0.04 | 0.08 | 24 |
| Mu | 0.38 | 0.12 | 0.19 | 24 |
| Nu | 0.71 | 0.42 | 0.53 | 24 |
| Omega | 0.26 | 0.62 | 0.37 | 24 |
| Omicron | 0.34 | 0.75 | 0.47 | 24 |
| Phi | 0.19 | 0.12 | 0.15 | 24 |
| Pi | 0.59 | 0.42 | 0.49 | 24 |
| Psi | 0.00 | 0.00 | 0.00 | 22 |
| Rho | 0.83 | 0.21 | 0.33 | 24 |
| Sigma | 0.28 | 0.29 | 0.29 | 24 |
| Tau | 0.72 | 0.54 | 0.62 | 24 |
| Theta | 0.21 | 0.54 | 0.31 | 24 |
| Upsilon | 0.00 | 0.00 | 0.00 | 24 |
| Xi | 0.39 | 0.62 | 0.48 | 24 |
| Zeta | 0.40 | 0.08 | 0.14 | 24 |
| Accuracy | | | 0.34 | 574 |
| Macro (avg) | 0.37 | 0.34 | 0.29 | 574 |
| Weighted (avg) | 0.37 | 0.34 | 0.29 | 574 |

quite close to Hell-Char and PaLit-Char Upsilon (similar to Y). The generalization task becomes an unguided extrapolation, resulting in low Accuracy.

# E TRANSFORMERS

In subsections 5.2 and 5.3, we presented CNN models for classification and clustering. In addition to these, we also trained a transformer-based model. Specifically, we fine-tuned the pre-trained Swin Vision Transformer (Liu et al., 2021) and achieved a classification accuracy of 0.84. However, applying SCL loss and Lacunae augmentation did not lead to further improvements. Nevertheless, as shown in Table 10, the clustering performance of the Swin+LF+DSCL model surpasses that of the plain Swin model, indicating that the embeddings are of higher quality despite no improvement in classification accuracy. Even this improved performance, however, falls behind that of our ResNet18+LF+DSCL across clustering algorithms and metrics (Table 2).

## E.1 BASELINES WITH GLOBAL DEPENDENCIES

Besides ResNet-18, we also experimented with ConvNeXt-V2, which employ layer normalisation, global response normalisation, and convolutional masked autoencoders. Furthermore, we experimented with ViT-16S, which uses self-attention instead of convolutional layers and captures global dependencies. Table 11 shows that F1 of both models increases when we add our LF and DSCL. As can be seen on Table 12, the best performance across different backbones is achieved by ResNet-18, using the Spectral clustering algorithm. This contradicts the better classification performance of ViT-16S and ConvNeXt-V2 (Table 11), which is likely due to overfitting.

Table 7: Classification performance of ConvNeXt-V2+LF+DSCL on Hell-Char.

| Class | Precision | Recall | F1-Score | Support |
|---|---|---|---|---|
| Alpha | 0.78 | 0.65 | 0.71 | 139 |
| Beta | 0.69 | 0.93 | 0.79 | 67 |
| Chi | 0.74 | 0.94 | 0.83 | 85 |
| Delta | 0.78 | 0.83 | 0.81 | 113 |
| Epsilon | 0.83 | 0.88 | 0.85 | 138 |
| Eta | 0.82 | 0.90 | 0.86 | 124 |
| Gamma | 0.82 | 0.81 | 0.81 | 105 |
| Iota | 0.86 | 0.84 | 0.85 | 141 |
| Kappa | 0.87 | 0.93 | 0.90 | 127 |
| Lambda | 0.83 | 0.65 | 0.73 | 117 |
| Mu | 0.87 | 0.94 | 0.90 | 126 |
| Nu | 0.88 | 0.92 | 0.90 | 134 |
| Omega | 0.88 | 0.87 | 0.87 | 126 |
| Omicron | 0.88 | 0.79 | 0.83 | 136 |
| Phi | 0.85 | 0.95 | 0.90 | 83 |
| Pi | 0.90 | 0.86 | 0.88 | 127 |
| Psi | 0.48 | 0.71 | 0.57 | 17 |
| Rho | 0.90 | 0.89 | 0.89 | 133 |
| Sigma | 0.92 | 0.91 | 0.91 | 138 |
| Tau | 0.94 | 0.85 | 0.89 | 139 |
| Theta | 0.91 | 0.90 | 0.90 | 86 |
| Upsilon | 0.95 | 0.80 | 0.87 | 133 |
| Xi | 0.87 | 0.87 | 0.87 | 47 |
| Zeta | 0.73 | 0.73 | 0.73 | 22 |
| Accuracy | | | 0.85 | 2603 |
| Macro (avg) | 0.83 | 0.85 | 0.84 | 2603 |
| Weighted (avg) | 0.86 | 0.85 | 0.85 | 2603 |

Table 8: Classification Performance on Hell-Char, per letter, of ViT-16S+LF+DSCL

| Class | Precision | Recall | F1-Score | Support |
|---|---|---|---|---|
| Alpha | 0.68 | 0.73 | 0.70 | 139 |
| Beta | 0.97 | 0.91 | 0.94 | 67 |
| Chi | 0.93 | 0.98 | 0.95 | 85 |
| Delta | 0.89 | 0.84 | 0.86 | 113 |
| Epsilon | 0.87 | 0.89 | 0.88 | 138 |
| Eta | 0.84 | 0.87 | 0.85 | 124 |
| Gamma | 0.83 | 0.75 | 0.79 | 105 |
| Iota | 0.84 | 0.84 | 0.84 | 141 |
| Kappa | 0.89 | 0.91 | 0.90 | 127 |
| Lambda | 0.80 | 0.65 | 0.72 | 117 |
| Mu | 0.84 | 0.90 | 0.87 | 126 |
| Nu | 0.90 | 0.84 | 0.87 | 134 |
| Omega | 0.91 | 0.87 | 0.89 | 126 |
| Omicron | 0.82 | 0.91 | 0.86 | 136 |
| Phi | 0.90 | 0.96 | 0.93 | 83 |
| Pi | 0.85 | 0.91 | 0.88 | 127 |
| Psi | 0.88 | 0.82 | 0.85 | 17 |
| Rho | 0.94 | 0.80 | 0.87 | 133 |
| Sigma | 0.88 | 0.90 | 0.89 | 138 |
| Tau | 0.86 | 0.81 | 0.83 | 139 |
| Theta | 0.88 | 0.81 | 0.84 | 86 |
| Upsilon | 0.79 | 0.90 | 0.85 | 133 |
| Xi | 0.94 | 0.94 | 0.94 | 47 |
| Zeta | 0.87 | 0.91 | 0.89 | 22 |
| Accuracy | | | 0.86 | 2603 |
| Macro (avg) | 0.87 | 0.86 | 0.86 | 2603 |
| Weighted (avg) | 0.86 | 0.86 | 0.86 | 2603 |

Table 9: Classification performance of ResNet18+LF+DSCL trained on Hell-char, fine-tuned on PaLit-char and tested on Med-Char

| Class | Precision | Recall | F1-score | Support |
|---|---|---|---|---|
| Alpha | 0.10 | 0.12 | 0.11 | 24 |
| Beta | 0.29 | 0.17 | 0.21 | 24 |
| Chi | 0.74 | 0.96 | 0.84 | 24 |
| Delta | 0.50 | 0.08 | 0.14 | 24 |
| Epsilon | 0.49 | 0.79 | 0.60 | 24 |
| Eta | 0.86 | 0.25 | 0.39 | 24 |
| Gamma | 0.00 | 0.00 | 0.00 | 24 |
| Iota | 0.68 | 0.88 | 0.76 | 24 |
| Kappa | 0.81 | 0.54 | 0.65 | 24 |
| Lambda | 1.00 | 0.38 | 0.55 | 24 |
| Mu | 0.62 | 0.62 | 0.62 | 24 |
| Nu | 0.33 | 0.04 | 0.07 | 24 |
| Omega | 0.25 | 0.83 | 0.38 | 24 |
| Omicron | 0.37 | 0.96 | 0.53 | 24 |
| Phi | 0.73 | 0.79 | 0.76 | 24 |
| Pi | 0.44 | 0.17 | 0.24 | 24 |
| Psi | 0.75 | 0.55 | 0.63 | 22 |
| Rho | 0.83 | 0.62 | 0.71 | 24 |
| Sigma | 0.48 | 0.50 | 0.49 | 24 |
| Tau | 0.55 | 0.71 | 0.62 | 24 |
| Theta | 0.44 | 0.67 | 0.53 | 24 |
| Upsilon | 0.03 | 0.04 | 0.04 | 24 |
| Xi | 0.62 | 0.75 | 0.68 | 24 |
| Zeta | 1.00 | 0.17 | 0.29 | 24 |
| Accuracy | | | 0.48 | 574 |
| Macro (avg) | 0.54 | 0.48 | 0.45 | 574 |
| Weighted (avg) | 0.54 | 0.48 | 0.45 | 574 |

Table 10: Clustering performance using different configurations of the SWIN architecture

| | k-means | | Spectral | | AH | |
|---|---|---|---|---|---|---|
| Embedding | NMI | ARI | NMI | ARI | NMI | ARI |
| SWIN+LF+DSCL | **0.633** | **0.404** | **0.785** | **0.700** | **0.772** | **0.690** |
| SWIN | 0.449 | 0.243 | 0.595 | 0.395 | 0.575 | 0.390 |

Table 11: Classification performance on Hell-Char (sorted) of ViT-16S and ConvNeXt-V2, pre-trained (PT) and fine-tuned (FT), when we add our DSCL and LF.

| Model | Fragmentation | Contrastive Loss | Accuracy | F1 |
|---|---|---|---|---|
| ConvNeXt-V2 | – | – | 0.848 | 0.836 |
| ConvNeXt-V2 | LF | DSCL | **0.851** | **0.854** |
| ViT-16S | – | – | **0.867** | 0.840 |
| ViT-16S | LF | DSCL | 0.856 | **0.850** |

Table 12: Clustering performance of different backbones using LF and DSCL, on Hell-Char, sorted by performance of the (best-performing) Spectral algorithm.

| | k-means | | Spectral | | AH | |
|---|---|---|---|---|---|---|
| Embedding | NMI | ARI | NMI | ARI | NMI | ARI |
| ResNet18+LF+DSCL | 0.667 | 0.411 | **0.836** | **0.743** | **0.818** | **0.726** |
| ViT-16S+LF+DSCL | **0.796** | **0.714** | 0.802 | 0.727 | 0.787 | 0.688 |
| ConvNeXt-V2+LF+DSCL | 0.776 | 0.683 | 0.786 | 0.674 | 0.755 | 0.626 |
| Swin+LF+DSCL | 0.633 | 0.404 | 0.785 | 0.700 | 0.772 | 0.690 |

