# OpenReview forum: "Representation Learning of Ancient Greek Letterforms across Time"
_ICLR.cc/2026/Conference — Submitted to ICLR 2026_

### Official Review · Reviewer_KveJ · 2025-10-15

**Soundness:** 2
**Presentation:** 3
**Contribution:** 2
**Rating:** 2
**Confidence:** 3

**Summary:**

This paper addresses the challenge of diachronic representation learning for ancient Greek handwriting by introducing three datasets. Temporal generalization tests show strong performance on PaLit-Char (84% accuracy) but limited performance on Med-Char.

**Strengths:**

1.The three cross-temporal datasets fill a gap in ancient Greek handwriting research, providing a standardized benchmark for low-resource, diachronic visual recognition.

**Weaknesses:**

1.The core methodological components (supervised contrastive loss, data augmentation) are adaptations of existing frameworks (Khosla et al., 2020 for SCL; general image augmentation for corruption simulation) rather than novel theoretical constructs. The "similarity weighting" and "lacuna-driven" modifications are incremental tweaks to fit the ancient manuscript domain, without introducing new mathematical formulations, learning paradigms, or theoretical insights that advance the broader field of representation learning—this limits the work’s contribution beyond domain-specific application.

2.The sharp accuracy drop on Med-Char (45%) is attributed to letterform change but lacks exploration of solutions (e.g., cross-dataset fine-tuning).

3.The paper fails to specify key technical details: how the dynamic similarity matrix in the contrastive loss is computed (global vs. batch-wise) and update frequency.

4.The paper does not clarify parameters for lacuna augmentation (e.g., size, density, curvature of lacunae) or validate their impact on model performance.

5.It Only tests Swin Transformer among Transformer models, with no exploration of variants (e.g., ViT, DeiT) that may better capture local stroke features.

**Questions:**

1.The authors should explicitly clarify the theoretical novelty of the similarity-weighted supervised contrastive loss. How does it differ from existing SCL variants beyond adding a domain-specific weight term? Are there new theoretical insights derived from this modification?

2.How is the dynamic similarity matrix in the contrastive loss calculated? Is it based on the entire training set or individual batches?

3.What criteria determined the parameters (shape, size) of lacunae in the augmentation? Were they informed by paleographic studies of manuscript degradation?

4.Why was only Swin Transformer tested? Have you considered Transformer variants optimized for local features (e.g., ConvNeXt-V2) for better performance?

---

> ### Author Response · Authors · 2025-11-26
> **Weakness: Adaptations instead of constructs**
>
> ```The core methodological components (supervised contrastive loss, data augmentation) are **adaptations of existing frameworks** (Khosla et al., 2020 for SCL; general image augmentation for corruption simulation) rather than novel theoretical constructs. The "similarity weighting" and "lacuna-driven" modifications are **incremental tweaks** to fit the ancient manuscript domain, without introducing new mathematical formulations, learning paradigms, or theoretical insights that advance the broader field of representation learning—this limits the work’s contribution beyond domain-specific application.```
>
> ---
>
> While we agree that our approach builds on the established Supervised Contrastive Learning (SCL) and Data Augmentation methods, we assert that our Similarity-Weighted (SW) SCL and Lacuna-driven Augmentation are **not mere “incremental tweaks”, but rather domain-aware innovations** that benefit the field of paleography and are a great example of representation learning under challenging data conditions aligning with domain-specific adaptations seen in other publications.
>
> To better give the context of our contribution, we also draw parallels to related work presented at ICLR. SW-SCL is a general approach and consistent with the recognised advance presented in [1], which proves that similarity-based weighting and mining pairs of non-identical samples with similar characteristics for contrastive learning is a significant methodological contribution that advances sample selection and loss formulation across diverse deep learning domains, not just a minor tweak. Our approach is also conceptually similar to the objective of another study [2], which aims to improve soft similarity to outperform existing contrastive methods. Our lacuna-driven augmentation is a domain-informed strategy that faithfully simulates realistic manuscript degradations. This mirrors the contribution of [3], which demonstrates that forming views by corrupting a random subset of features (i.e., domain-specific augmentation/corruption) is a widely-applicable, simple and effective technique to advance contrastive learning, even on non-vision tasks like tabular data.
>
> [1] Nie Lin, Takehiko Ohkawa, Mingfang Zhang, Yifei Huang, Minjie Cai, Ming Li, Ryosuke Furuta, and Yoichi Sato, “SiMHand: Mining Similar Hands for Large-Scale 3D Hand Pose Pre-training,” ICLR 2025.
>
> [2] Sobal, Vlad, Mark Ibrahim, Randall Balestriero, Vivien Cabannes, Diane Bouchacourt, Pietro Astolfi, Kyunghyun Cho, and Yann LeCun. "$\mathbb {X} $-Sample Contrastive Loss: Improving Contrastive Learning with Sample Similarity Graphs," ICLR 2026.
>
> [3] Bahri, Dara, Heinrich Jiang, Yi Tay, and Donald Metzler. "Scarf: Self-supervised contrastive learning using random feature corruption," ICLR 2022.

---

> ### Author Response · Authors · 2025-11-26
> **Weakness: Lacking evidence**
>
> ```The sharp accuracy drop on Med-Char (45%) is attributed to letterform change but lacks exploration of solutions (e.g., cross-dataset fine-tuning).```
>
> ---
>
> We fine-tuned ResNet+Lacuna+SWSCL on PaLit and then applied it to MedChar, leading to no improvements (macro-averaged F1 of 45%). We will add this result to the paper.

---

> ### Author Response · Authors · 2025-11-26
> **Weakness: Lacking details**
>
> ```The paper fails to specify key technical details: how the dynamic similarity matrix in the contrastive loss is computed (global vs. batch-wise) and update frequency.```
>
> ---
>
> Similarity-weighting in supervised contrastive loss (SCL) is based on the global representations that are actively learned. We re-estimate the class-similarity matrix every 3 epochs. At each update, we pass the entire training set through the current model, we compute class prototypes from the normalised embeddings, and we derive the cosine similarity between prototypes (diagonal entries set to zero). This yields a dynamic measure of inter-class similarity that evolves with the representation space. The updated matrix is then used by our Similarity-Weighted (SW) SCL, which down-weights negatives from highly similar classes and up-weights negatives from dissimilar ones. **We will add this to the paper.**
>
> ---
>
> ```The paper does not clarify parameters for lacuna augmentation (e.g., size, density, curvature of lacunae) or validate their impact on model performance.```
>
> ---
>
> For each image, we sample 1–4 lacunae and each covers 2–15% of the area to match the typical size distribution of physical papyrus damage. Lacuna geometry is obtained by drawing anisotropic ellipses whose contours are further distorted via random morphological operations (erosion or dilation), producing organic, non-rectangular shapes characteristic of flaking, humidity damage, parchment wear, or insect deterioration (e.g., worm holes are frequent in papyri). These lacunae are placed at random positions and the masked pixels are replaced with background values, reflecting the absence of ink/papyrus rather than additive noise. This augmentation increases robustness to fragmentary handwriting and introduces realistic variability at negligible computational cost. **We will add this to the paper.**

---

> ### Author Response · Authors · 2025-11-26
> **Weakness: Lacking baselines**
>
> This is a fair point, which we addressed by adding ViT and ConvNext-V2. As can be seen in the table below, adding our Lacuna-based augmentation and similarity-weighted (SW) supervised contrastive loss (SCL) improves both. This result directly improves our work, and we will incorporate it in the paper. We will also update Figure 4 based on this.
>
> | Model / Configuration | Kmeans-NMI | Kmeans-ARI | Classification-F1 |
> | :--- | :---: | :---: | :---: |
> | ConvNext-V2 | 0.779 | 0.680 | 0.836 |
> |       +Lacuna +SW-SCL (ConvNext-V2) | **0.795** | **0.710** | **0.854** |
> | ViT+SCL | 0.800 | 0.725 | 0.840 |
> |   +Lacuna +SW-SCL (ViT+SCL) | **0.805** | **0.727** | **0.850** |

---

> ### Author Response · Authors · 2025-11-26
> **Question Answering**
>
> ```How is the dynamic similarity matrix in the contrastive loss calculated? Is it based on the entire training set or individual batches?```
>
> ---
>
> Similarity-weighting in SCL is based on the global representations that are actively learned. We re-estimate the class-similarity matrix periodically (i.e., every 3 training epochs). At each update, we pass the entire training set through the current model, we compute class prototypes from the normalised embeddings, and we derive the cosine similarity between prototypes (diagonal entries are set to zero). This yields a dynamic measure of inter-class similarity that evolves with the representation space. The updated matrix is then used by our Similarity-Weighted (SW) SCL, which down-weights negatives from highly similar classes and up-weights negatives from dissimilar ones.
>
> ---
>
> ```The authors should explicitly clarify the theoretical novelty of the similarity-weighted supervised contrastive loss. How does it differ from existing SCL variants beyond adding a domain-specific weight term? Are there new theoretical insights derived from this modification?```
>
> ---
>
> Standard SCL treats negatives uniformly, making classes with inherent affinities (e.g., letters with similar shapes, as psi-phi) to be strongly repelled. This leads to embeddings that fail to reflect natural inter-class relationships (see the last paragraph of Section 2). Our results (expanded to address a raised weakness, see our response to _lacking baselines_) reflect the superiority of our approach. We also note that engineering a similarity matrix is time consuming and yields worse results (noted in Footnote 1).
>
> ---
>
> ```What criteria determined the parameters (shape, size) of lacunae in the augmentation? Were they informed by paleographic studies of manuscript degradation?```
>
> ---
>
> Yes, we elaborated in our response for the respective weakness that was correctly raised. We will clarify this in the paper.
>
> ---
>
> ```Why was only Swin Transformer tested? Have you considered Transformer variants optimized for local features (e.g., ConvNeXt-V2) for better performance?```
>
> ---
>
> We agree, so we incorporated two more backbones. The results, presented in our response for the respective weakness, show that both improve when we add our lacuna-driven augmentation and SW-SCL.

---

### Official Review · Reviewer_81q8 · 2025-10-27

**Soundness:** 2
**Presentation:** 3
**Contribution:** 2
**Rating:** 2
**Confidence:** 3

**Summary:**

This paper addresses learning representations for ancient Greek letter forms as attested in ancient manuscripts across eras and handwriting styles. The paper curates a set of ancient Greek handwriting datasets, proposes modifications to a representation learning scheme to improve performance on ancient Greek handwriting, and shows results of paleographic analysis aided by these representations.

**Strengths:**

Computational paleography is an important and under-explored field, and it is encouraging to see modern representation learning methods being leveraged to aid expert analysis.

The paper curates data for studying ancient Greek paleography with computational methods from disparate sources, making them more accessible for future work.

It appears that the learned representations with the proposed method are more effective for paleographic analysis and yield valuable insights.

The anonymous code is appreciated for aiding transparency and reproducibility.

**Weaknesses:**

The technical contribution of the paper appears limited, both regarding methodology and data.

The components proposed as novel are (A) similarity-weighted supervised contrastive loss and (B) lacuna augmentation. However, neither of these are fully explained, and both have conceptual issues which are not addressed:

* (A) Similarity-weighted supervised contrastive loss: The term S_{y_i, y_a} is given as “dynamically computed from embeddings”. It is not clear how it is computed, whether these embeddings are the representations actively being learned, or if these refer to fixed, pre-computed embeddings. Conceptually, the contrastive learning objective is already expected to model visual similarity between samples, so it is unclear what motivates adding an additional similarity term. In addition, L155 mentions an addition “standard cross-entropy” loss, but it is not clear if this refers to training the backbone on letter classification, and if so, whether the contrastive loss is applied to the output of the classification head or to intermediate activations.

* (B) Lacuna augmentation: While this is presented as one of the main contributions of the paper, it is never clearly defined. L150 mentions that this augments images by masking; if so, this is similar to standard image augmentation strategies for training vision models.

There also lacks a comparison to stronger representation learning baselines such as zero-shot or fine-tuned CLIP, DINO, or DIFT features, while comparing only to baselines like pretrained ResNet features which are expected to be weak on challenging images.

While the curation and aggregation of data is a valuable contribution, much of the data is sourced from existing datasets (Hell-Date, PaLit), and it is unclear how much new data is contributed. The abstract and intro do not mention this, suggesting that this is a new data contribution. It should be clarified whether new data is being contributed, or whether the main contribution is the curation or filtering of existing data.

Overall, the bulk of the paper is devoted to exploratory data analysis and paleographic insights from methods such as clustering. While these are valuable insights in general, it is not clear whether they are in the scope of this conference.

**Questions:**

Is the proposed method specific to ancient Greek? It seems like the proposed method could be applied more generally.

Is Med-Char sourced from an existing dataset, or new data being released? Similarly, as suggested on L333, do the other datasets being contributed contain new data?

When you discuss “PCA” or “Otsu+PCA” features (Sec 5.2), does this refer to PCA applied directly to pixel intensity values?

---

> ### Author Response · Authors · 2025-11-26
> **Weakness: Novel components are not explained and have conceptual issues**
>
> ```Similarity-weighted supervised contrastive loss [...] not clear how it is computed, whether these embeddings are the representations actively being learned, or if these refer to fixed, pre-computed embeddings.```
>
> ---
>
> Similarity-weighting in supervised contrastive loss (SCL) is based on the global representations that are actively learned. Specifically, we re-estimate the class-similarity matrix periodically (i.e., every 3 training epochs). At each update, we pass the entire training set through the current model, we compute class prototypes from the normalised embeddings, and we derive the cosine similarity between prototypes (diagonal entries are set to zero). This yields a dynamic measure of inter-class similarity that evolves with the representation space. The updated matrix is then used by our Similarity-Weighted (SW) SCL, which down-weights negatives from highly similar classes and up-weights negatives from dissimilar ones. **We will clarify this.**
>
> ---
>
> ``` Conceptually, the contrastive learning objective is already expected to model visual similarity between samples, so it is unclear what motivates adding an additional similarity term.```
>
> ---
>
> Standard SCL treats negatives uniformly, making classes with inherent affinities (e.g., letters with similar shapes, as psi-phi) to be strongly repelled. This leads to embeddings that fail to reflect natural inter-class relationships (see the last paragraph of Section 2). Our results (expanded in response to the fair point about _lacking baselines_) reflect the superiority of our approach. We also note, however, that engineering a similarity matrix is not only time consuming, but our experiments show that it yields worse results (noted in Footnote 1). **We will clarify this.**
>
> ---
>
> ```In addition, L155 mentions an addition “standard cross-entropy” loss, but it is not clear if this refers to training the backbone on letter classification, and if so, whether the contrastive loss is applied to the output of the classification head or to intermediate activations.```
>
> ---
>
> The standard cross-entropy loss refers to the supervised letter-classification objective applied to the backbone’s classification head. The contrastive loss is not computed on the classification logits; it is applied to the intermediate feature embeddings produced by the backbone before the classification layer. Thus, the model jointly optimises cross-entropy on the classification head and contrastive loss on the shared backbone representations.

---

> ### Author Response · Authors · 2025-11-26
> **Weakness: Lacuna augmentation - unclear definition**
>
> ```(B) Lacuna augmentation: While this is presented as one of the main contributions of the paper, it is never clearly defined. L150 mentions that this augments images by masking; if so, this is similar to standard image augmentation strategies for training vision models.```
>
> ---
>
> Synthetic fragmentation using elliptical fragments is a task-driven augmentation strategy that is better suited to our fragmented material. It attempts to simulate manuscript deterioration more realistically than standard erasure augmentations by inserting irregular regions that approximate actual lacunae observed in historical documents. For each image, we sample 1–4 lacunae and each covers 2–15% of the area to match the typical size distribution of physical papyrus damage. Lacuna geometry is obtained by drawing anisotropic ellipses whose contours are further distorted via random morphological operations (erosion or dilation), producing organic, non-rectangular shapes characteristic of flaking, humidity damage, parchment wear, or insect deterioration (e.g., worm holes are frequent in papyri). These lacunae are placed at random positions and the masked pixels are replaced with background values, reflecting the absence of ink/papyrus rather than additive noise. This augmentation increases robustness to fragmentary handwriting and introduces realistic variability (at negligible computational cost). Several vision tasks that involve machine learning for ancient languages (even recent, e.g., https://www.nature.com/articles/s41586-025-09292-5) could benefit from such an approach.

---

> ### Author Response · Authors · 2025-11-26
> **Weakness: Lacking comparison to baselines**
>
> This is a fair point, which we addressed by adding ViT and ConvNext-V2 (requested by another reviewer). As can be seen in the table below, adding our Lacuna-based augmentation and similarity-weighted (SW) supervised contrastive loss (SCL) improves both. This result directly improves our work, and we will incorporate it in the paper. We will also update Figure 4 based on this.
>
> | Model / Configuration | Kmeans-NMI | Kmeans-ARI | Classification-F1 |
> | :--- | :---: | :---: | :---: |
> | ConvNext-V2 | 0.779 | 0.680 | 0.836 |
> |       +Lacuna +SW-SCL (ConvNext-V2) | **0.795** | **0.710** | **0.854** |
> | ViT+SCL | 0.800 | 0.725 | 0.840 |
> |   +Lacuna +SW-SCL (ViT+SCL) | **0.805** | **0.727** | **0.850** |

---

> ### Author Response · Authors · 2025-11-26
> **Weakness: Data novelty**
>
> ```much of the data is sourced from existing datasets (Hell-Date, PaLit), and it is unclear how much new data is contributed. The abstract and intro do not mention this, suggesting that this is a new data contribution. It should be clarified whether new data is being contributed, or whether the main contribution is the curation or filtering of existing data.```
>
> ---
>
> Med-Char and PaLit-Char are new datasets, compiled for this paper, which will be released after publication. Hell-Char is a subset of an existing dataset called Hell-Date. The data was already released in the past, but never used for applications as far as we know. **We will add this information.**

---

> ### Author Response · Authors · 2025-11-26
> **Answers to questions**
>
> ```Is the proposed method specific to ancient Greek? It seems like the proposed method could be applied more generally. ```
>
> ---
>
> The method can be applied generally, but it is expected to be particularly useful for poorly sourced languages, such as Ancient Greek, Tocaric, etc. We will clarify this.
>
> ---
>
> ```When you discuss “PCA” or “Otsu+PCA” features (Sec 5.2), does this refer to PCA applied directly to pixel intensity values?```
>
> ---
>
> Characters have consistent alignment and size, hence pixel-based variance captured by PCA can correspond to meaningful features of the characters (e.g., strokes and overall shape). Although it destroys 2D structure (edges, texture) and does not focus on separability, we applied PCA to the raw input as a simple preprocessing baseline used in literature [1], which is complementary to CNN features that preserve local structure.
>
> [1] Paparigopoulou et al. (2022) "Dating Greek papyri images with machine learning." ICDAR Workshop on Computational Paleography.

---

### Official Review · Reviewer_fckt · 2025-10-28

**Soundness:** 2
**Presentation:** 3
**Contribution:** 2
**Rating:** 4
**Confidence:** 3

**Summary:**

This paper proposes a method for representation learning of ancient Greek letterforms across time, using three historical datasets (Hell-Char, PaLit-Char, and Med-Char). The method combines similarity-weighted supervised contrastive loss with a lacunae enhancement strategy to improve performance in classification, clustering, and prototype visualization. Experimental results show that the proposed method outperforms baseline approaches in terms of accuracy and clustering quality.

**Strengths:**

1. The method applies contrastive learning and data augmentation to the problem of historical text recognition, offering potential practical value.

2. The results show the effectiveness of the proposed method over baselines in classification and clustering tasks.

3. The dataset and methodology could be valuable for future historical document analysis tasks.

**Weaknesses:**

1.  The method builds on established techniques (contrastive learning, supervised contrastive loss, and data augmentation). The primary innovation lies in its application to historical text recognition rather than new methodological contributions.

2. The paper lacks detailed descriptions of how the similarity matrix is updated and how lacunae enhancement is applied.

3. The approach uses circular or elliptical shapes for missing characters (lacunae), but the paper does not provide enough detail on how these are generated.

**Questions:**

1. Could you clarify the novel methodological contributions beyond the application of existing techniques to a new domain? This would help us better understand the innovation in your approach.

2. Could you provide more details on the implementation of the lacunae enhancement strategy?

3. How did you prevent data leakage, particularly ensuring that the same document or scribe's handwriting is not repeated in both the training and validation sets?

---

> ### Author Response · Authors · 2025-11-25
> **Weakness: innovation lies in application**
>
> ```The method builds on established techniques (contrastive learning, supervised contrastive loss, and data augmentation). The primary innovation lies in its application to historical text recognition rather than new methodological contributions.```
> ---
>
> Although based on observations drawn from a specific field, our two contributions are methodological that move beyond the established techniques and which improve all backbone models (we added two more).
>
> __Synthetic fragmentation__ (Random Lacunae) simulates manuscript deterioration more realistically than standard erasure augmentations using elliptical fragments. This strategy is particularly suited to our material, which also has fragments but several vision tasks that involve machine learning for ancient languages (even recent, e.g., https://www.nature.com/articles/s41586-025-09292-5) could benefit from such an approach.
>
> __Dynamically similarity weighting in supervised contrastive loss (SCL)__, on the other hand, offers interpretability of class similarity and helps the model. By updating the similarity matrix, the model’s representation (of which classes look alike) evolves during training, keeping it in sync and ensuring that the contrastive loss will always focus on the most confusable classes. Standard SCL treats negatives uniformly and classes with inherent affinities (e.g., letters with similar shapes, as psi-phi) are strongly repelled, yielding embeddings that fail to reflect natural inter-class relationships (see the last paragraph of Section 2). Engineering a similarity matrix could be an alternative, but our experiments gave worse results (see Footnote 1).

---

> ### Author Response · Authors · 2025-11-25
> **Weakness: Lacking details**
>
> ```The paper lacks detailed descriptions of how the similarity matrix is updated```
> ---
> We re-estimate the class-similarity matrix periodically during training (every 3 epochs). At each update, we pass the entire training set through the current model, compute class prototypes from the normalized embeddings, and derive the cosine similarity between prototypes (diagonal entries are set to zero). This yields a dynamic measure of inter-class similarity that evolves with the representation space. (An exponential moving average can be applied to stabilize updates.) The updated matrix is then used by our Similarity-Weighted Supervised Contrastive Loss, which down-weights negatives from highly similar classes and up-weights negatives from dissimilar ones. **We will add this information.**
>
> ```The approach uses circular or elliptical shapes for missing characters (lacunae), but the paper does not provide enough detail on how these are generated.```
> ---
> RandomLacunae attempts to simulate manuscript deterioration more realistically than standard erasure augmentations by inserting irregular regions that approximate actual lacunae observed in historical documents. For each image, we sample 1–4 lacunae and each covers 2–15% of the area to match the typical size distribution of physical papyrus damage. Lacuna geometry is obtained by drawing anisotropic ellipses whose contours are further distorted via random morphological operations (erosion or dilation), producing organic, non-rectangular shapes characteristic of flaking, humidity damage, or parchment wear. These lacunae are placed at random positions and the masked pixels are replaced with background values, reflecting the absence of ink/papyrus rather than additive noise. This augmentation increases robustness to fragmentary handwriting and introduces realistic variability (at negligible computational cost). **We will add this information.**

---

> ### Author Response · Authors · 2025-11-25
> **Question: Data Leakage**
>
> _How did you prevent data leakage, particularly ensuring that the same document or scribe's handwriting is not repeated in both the training and validation sets?_
>
> ---
> We frame paleographic problems (dating/attribution) as the search for a strong script embedding, where classification relies on defining distance thresholds (same period/scribe). Our current work validates robust letter representations using classification. Testing against a chronologically distinct external dataset (Table 4) confirmed the absence of validation leakage (i.e., accuracy did not drop). While this validates the model's integrity, it confirms that our representations do not capture the fidelity required for scribal identification, which remains a challenging task, very hard to solve. This limitation is likely imposed by the low sample density (max 120 characters per papyrus). **We will add this observation.**

---

> ### Author Response · Authors · 2025-11-25
> **Question: Novel methodological contributions**
>
> _Could you clarify the **novel methodological contributions beyond the application** of existing techniques to a new domain? This would help us better understand the innovation in your approach._
>
> ---
>
> Erasing input as augmentation in image classification is not new [1] and it has been shown particularly useful for papyri, which are often fragmented (i.e., parts of the images are already missing). In this work, we present a synthetic augmentation that is closer in nature to the real fragments (i.e., elliptic instead of square) and a comparison between rows 2-3 of Table 1 shows that this approach is better.  Similarity-weighting in supervised contrastive loss, on the other hand, besides interpretability (representation class similarity), helps the model avoid confusion. In Figure 6 (of the Appendix), for example, only one (alpha-lambda) out of the four pairs of high similarity noted in the caption get a high value in the confusion matrix (Figure 5 of the Appendix). **We will clarify this.**
>
> [1]: Zhong, Z., Zheng, L., Kang, G., Li, S., & Yang, Y. (2017). Random Erasing Data Augmentation. arXiv preprint arXiv:1708.04896.

---

### Author Response · Authors · 2025-11-26
**Acknowledging the quality of the reviews**

We thank all the reviewers for their thoughtful points raised. We acknowledge that our work improved by addressing them.

In sum, we:
1. explained the novelty of our two methodological contributions, lacuna-augmentation (LA) and similarity-weighed (SW) SCL;
1. added two baselines, showing that they improve when we add our two technical innovations, LA and SW;
2. provided missing details on the implementations of LA and SW;
4. explained the novelty of our provided datasets.

We sincerely hope that our response mitigated successfully the raised concerns, leading to a score increase. But we can elaborate more, if needed.

---

> ### Author Response · Authors · 2025-12-02
> **Revision addressing all points submitted**
>
> Driven by our responses to the raised concerns, we revised our submission in the following ways:
>
> 1) We **highlighted our technical contribution** by adding experimental settings (5.1) and clarifications in the methodology about our proposed LF and DSCL, also adding a Discussion section (7) elaborating on our technical novelty.
>
> 2) we showed that **our LF and DSCL improve also ViT and ConvNeXt-V2,** the two backbones suggested by the reviewers (Table 11). Notably, however, they both underperform when compared to ResNet-18 in clustering (Table 12), which remains the best-performing model for our purposes (our Section 6 was kept intact).
>
> 3) We clarified that **two out of three datasets are novel** while the third has not been studied before. This directly increases the impact of our study.
>
> 4) We added an experiment on **cross-dataset fine-tuning,** showing that fine-tuning on PaLit-Char and testing on Med-Char is not leading to reasonable gains (Appendix D), and providing further context.
>
> Although all of these points were easy for us to address (i.e., cross dataset fine-tuning and adding baselines) or clarify (i.e., missing details and explanations), we acknowledge their impact and we consider that the current version addresses all the criticism previously raised.

---

### Meta-Review · Area_Chair_7MEJ · 2026-01-14

**Summary:**

Reviewers generally acknowledged the paper’s practical value (representation learning for Ancient Greek letterforms), the effort in dataset construction, and the method’s effectiveness on specific tasks. However, all three reviewers pointed out a **lack of substantial methodological novelty**: the proposed “similarity-weighted supervised contrastive loss” and “lacuna augmentation” were viewed as domain-specific adaptations of existing techniques (e.g., SCL, random erasing), rather than theoretical or algorithmic innovations. Additionally, reviewers repeatedly criticized the **insufficient technical details** (e.g., how the similarity matrix is updated, how lacuna parameters are chosen) and questioned the **rigor of experimental design** (e.g., inadequate evaluation of cross-era generalization, absence of stronger baselines like CLIP or DINO). Although the authors addressed some implementation details in the rebuttal and added experiments with ViT/ConvNeXt-V2, they failed to fundamentally overcome the perception of being an incremental contribution.

**Reviewer Concerns:**

| Concern | Addressed in Rebuttal? | Still Outstanding? |
| --- | --- | --- |
| **Methodological novelty is limited to domain adaptation** | Partially – Authors emphasized their approach as “domain-aware innovations” and cited ICLR papers with similar positioning, but offered no new theoretical framework or proof of generalizability. | **Yes** – Core components remain engineering adaptations of SCL and data augmentation, lacking broader contributions to representation learning. |
| **Insufficient technical details (similarity matrix, lacuna parameters)** | **Yes** – Authors clarified: similarity matrix is updated every 3 epochs using class prototypes computed from the full training set; lacunae consist of 1–4 ellipses covering 2–15% of image area, applied with morphological operations to mimic real damage. | No |
| **Lack of strong baselines (e.g., CLIP, DINO)** | Partially – Added ViT/ConvNeXt-V2 results, but did not include zero-shot CLIP or other cross-modal baselines explicitly requested. | **Yes** – Key baselines like CLIP/DIFT remain absent. |
| **Data novelty unclear (reuse of Hell-Date/PaLit)** | **Yes** – Clarified that Med-Char and PaLit-Char are newly constructed, while Hell-Char is a character-level subset of Hell-Date (first time used for character tasks). | No |
| **Poor generalization to Med-Char (45% acc) unaddressed** | **Yes** – Additional experiments confirmed that fine-tuning across datasets fails to improve performance (F1 remains ~45%). | No – but this result further highlights method limitations. |
| **Potential data leakage (same scribe in train/test)** | Partially – Authors used stable cross-era performance as indirect evidence, but did not describe explicit deduplication by scribe or manuscript. | **Yes** – No clear strategy for preventing train/test overlap at the document or scribe level was provided. |

**Reviewer Scores:**

+ **Reviewer fckt** (original score: 4 – marginally below acceptance):
Likely to increase to **4-6 (weak accept)**, as their main concerns (missing details, novelty clarification) were largely addressed, and authors demonstrated consistent gains across architectures.
+ **Reviewer 81q8** (original score: 2 – reject):
Likely to remain at **2–4 (reject to borderline)**, since core requests (comparison with CLIP, demonstration of foundational novelty) were unmet, and doubts about suitability for ICLR persist.
+ **Reviewer KveJ** (original score: 2 – reject):
Might increase slightly to **2-4 (borderline reject)** due to added technical clarifications and multi-architecture validation, but concerns about low Med-Char performance and weak theoretical contribution remain unresolved.

---

### Decision · Program_Chairs · 2026-01-26

Reject